# Bone Health in Children with Rheumatic Disorders: Focus on Molecular Mechanisms, Diagnosis, and Management

**DOI:** 10.3390/ijms23105725

**Published:** 2022-05-20

**Authors:** Francesca Di Marcello, Giulia Di Donato, Debora Mariarita d’Angelo, Luciana Breda, Francesco Chiarelli

**Affiliations:** Department of Pediatrics, University of Chieti, 66100 Chieti, Italy; francescadimarcello@gmail.com (F.D.M.); didonatogiuls@gmail.com (G.D.D.); deboramrdangelo@gmail.com (D.M.d.); luciana.bredach@gmail.com (L.B.)

**Keywords:** bone health, cartilage-bone interaction, secondary osteoporosis, rheumatic diseases, glucocorticoids, children

## Abstract

Bone is an extremely dynamic and adaptive tissue, whose metabolism and homeostasis is influenced by many different hormonal, mechanical, nutritional, immunological and pharmacological stimuli. Genetic factors significantly affect bone health, through their influence on bone cells function, cartilage quality, calcium and vitamin D homeostasis, sex hormone metabolism and pubertal timing. In addition, optimal nutrition and physical activity contribute to bone mass acquisition in the growing age. All these factors influence the attainment of peak bone mass, a critical determinant of bone health and fracture risk in adulthood. Secondary osteoporosis is an important issue of clinical care in children with acute and chronic diseases. Systemic autoimmune disorders, like juvenile idiopathic arthritis, can affect the skeletal system, causing reduced bone mineral density and high risk of fragility fractures during childhood. In these patients, multiple factors contribute to reduce bone strength, including systemic inflammation with elevated cytokines, reduced physical activity, malabsorption and nutritional deficiency, inadequate daily calcium and vitamin D intake, use of glucocorticoids, poor growth and pubertal delay. In juvenile arthritis, osteoporosis is more prominent at the femoral neck and radius compared to the lumbar spine. Nevertheless, vertebral fractures are an important, often asymptomatic manifestation, especially in glucocorticoid-treated patients. A standardized diagnostic approach to the musculoskeletal system, including prophylaxis, therapy and follow up, is therefore mandatory in at risk children. Here we discuss the molecular mechanisms involved in skeletal homeostasis and the influence of inflammation and chronic disease on bone metabolism.

## 1. Introduction

Bone tissue has many physiological functions: It provides protection to internal organs and allows movement; it contains bone marrow and offers an environment for the development of haematopoietic cells; finally, it acts as a reservoir of calcium and phosphate. Thanks to recent advances in molecular biology, the traditional concept of a passive tissue responding to hormonal and dietary influences has changed over time [1]. It is known by now that bone is a dynamic, adaptive tissue, responding to many different stimuli (hormonal, mechanical, pharmacological, and immunological). Approximately 40–90% of bone mineral composition variation is genetically determined [2]. Genetic factors affect bone metabolism through the influence on bone cell function, cartilage collagen quality and function, calcium and vitamin D homeostasis, sex hormone metabolism, and pubertal timing. In addition, environmental factors influence bone mass acquisition during growth. Optimal nutrition, physical activity and pubertal development play pivotal roles in bone health [3]. Furthermore, the strict interaction between bone and the immune system has been extensively investigated. Immune cells and haematopoietic stem cells (HSCs) are formed and maintained in the bone marrow, where they interact with bone cells [4]. Consequently, a key comorbidity of childhood rheumatic diseases is the potential for a negative impact on the growing skeletal development and bone metabolism. Skeletal assessment is therefore an important outcome in children with chronic inflammatory diseases [5]. In this review, we summarize the most important knowledge about bone metabolism and its secondary alterations in childhood immune-mediated diseases.

## 2. Bone Metabolism in Children

### 2.1. Joint Homeostasis and Cartilage–Bone Interaction

Articular cartilage plays an important role in joint homeostasis. It covers sub-chondral bone, facilitating relatively frictionless motion and absorbing the compressive forces generated by weight-bearing activities on bone itself. Articular cartilage is an avascular and alymphatic tissue, composed of chondrocytes within an extracellular matrix (ECM) containing water, collagen fibres, proteoglycans, and structural and regulatory proteins. A pericellular region immediately surrounding the chondrocytes, the pericellular matrix (PCM), is composed of type VI collagen and proteoglycans, providing an enclosed microenvironment in which optimal mechanical stress drives chondrogenesis. Chondrocytes are well-specialized cells that can sense and respond to mechanical stimuli through a mechanotransduction process. PCM absorbs dynamic and static forces and transmits them to the chondrocyte surface. These physical signals are sensed by calcium receptors and by molecules located in the cell membrane, such as channels, the primary cilium, and integrins [6,7,8,9]. Recently, the Piezo channels have shown key roles in mediating mechanotransduction in bone cells and are involved in bone disease. Piezo channels are mechanosensitive ion channels located in the cell membrane; they convert external mechanical stimuli into internal electrochemical signals in order to adapt to the cellular microenvironment. The electrochemical signal causes a series of intracellular, downstream signalling pathways. Piezo1 and Piezo2 are expressed in chondrocytes and participate in the maintenance of cartilage homeostasis. Piezo1 is expressed in bone and is responsible for mechanotransduction there [10].

Joint homeostasis and function depend on the mechanical and biological integrity of the components (chondrocytes and sub-chondral bone) and signalling exchanges between them. At the joint surface, superficial chondrocytes are flattened and placed parallel to joint surfaces; they produce a matrix rich in hyaluronan and lubricin. Middle zone chondrocytes are larger and rounder; in this zone, type II collagen fibrils increase in thickness. In the deeper zone, the chondrocytes are aligned in typical columns, which are perpendicular to the articular surface; this layer has the highest level of proteoglycans and the lowest water content [11]. The sub-chondral bone can be divided into several components, each of which performs specific functions. The sub-chondral plate, which is in direct contact with articular cartilage, transforms shear forces in compressive and tensile forces, avoiding damage to joint tissues. Below the sub-chondral plate, the bone is composed of a porous network of individual trabeculae oriented along “stress-lines” [12]. The coordinated action of bone cells (osteoblasts, osteoclasts, and osteocytes) makes this structure simultaneously highly sensitive and resistant to loads [13]. Sub-chondral bone contains vessels and channels, through which nutrients are transported to the bone and, potentially, to the deeper layers of cartilage; they also represent pathways for signalling molecules and factors across the osteochondral interface. During osteoarthritis (OA), new blood vessels are formed in the deep layer of articular cartilage, creating channels between the tissues. The newly formed channels increase the transport across the bone–cartilage interface, enhancing the crosstalk at the osteochondral interface [14,15]. Despite this, the primary route for nutrients to access the chondrocytes from sub-chondral bone is via diffusion [16]. Under physiological conditions, the density and orientation of collagen fibres in ECM can affect the diffusion of large molecules. Small molecules can diffuse easily through the osteochondral interface; however, this transport is altered by age and disease, while mechanical loading can increase the diffusive transport of molecules [17]. Transforming growth factor-beta (TGF-ß) and Wnt proteins (a family of secretory glycoproteins) are two important groups of molecules that play a role in both the maintenance and degradation of cartilage as they can diffuse across the osteochondral interface. Specifically, deregulation of TGF- ß and Wnt signalling causes instability of chondrocyte structure and function, altering their development and contributing to OA progression [18]. The TGF family are multipurpose growth factors that play a fundamental role in cartilage development, homeostasis, and repair. During the physiological loading of healthy joints, TGF-ß binds activin-like kinase receptors (ALK5 or ALK1), activating SMAD2/SMAD3 (mothers against decapentaplegic homolog 2 and 3) phosphorylation. This interaction contributes to chondrocyte metabolism and survival [19]. This action is also carried out by controlling inflammatory cytokine production. The pro-inflammatory cytokines, interleukin (IL)-1ß and tumour necrosis factor-α (TNF-α), released in joint tissue, are potent inducers of matrix metalloproteinases (MMPs) responsible for ECM components’ cleavage [20]. Wnt binds to its receptors, Frizzled (Fz), and to low-density lipoprotein receptor-related protein 5/6 (LRP5/6), activating β-catenin synthesis, which accumulates in the cell nucleus for use in gene transcription. In the absence of Wnt, β-catenin is degraded and not translocated to the cell nucleus [21]. There is a fine balance between the level of Wnt–Fz activation and the accumulation of β-catenin in the nucleus. Both excessive and insufficient Wnt activation, as well as increased and reduced β-catenin synthesis, can result in cartilage and sub-chondral bone damage [22]. Figure 1 summarizes the cartilage–bone interaction in the osteochondral unit.

### 2.2. Bone Tissue Remodeling and Peak Bone Mass

Bone tissue is a dynamic and highly specialized connective tissue; it provides a mechanical support for muscles and physical protection to the internal organs, as well as acting as a repository for minerals and haematopoietic cells. It comprises bone cells and a matrix composed of the organic materials of collagen (10%); proteins, proteoglycans and water (25%), which impart flexibility; and inorganic phosphate and calcium mineral salts (65%), which provide firmness [23]. Bone mineral deposition begins during pregnancy, and bone mineral content (BMC) increases 40-fold from birth until adulthood. The skeletal mass increases from approximately 70–95 g at birth to 2.400–3.300 g in young women and men [24]. Bone mineral density (BMD), the amount of bone mineral per square centimetre of bone, is genetically determined in 80% of cases. Genes play a key role in skeletal growth, starting from embryogenesis [25]. Bone mass increase also depends on environmental factors, such as physical activity, dietary calcium, adequate serum levels of vitamin D, maintenance of a healthy body weight, and hormonal status [26]. Bone is an active tissue, both in adults and in children. From birth until adulthood, it undergoes a constant process of modelling, which is prevalent in the developmental age, and remodelling, which is typical of adulthood. Remodelling is a process where osteoclasts and osteoblasts work sequentially in the same bone remodelling unit. Bone modelling describes the process whereby bones are shaped or reshaped by the independent action of osteoblasts and osteoclasts. Bone modelling results in skeletal development and growth and contributes to the periosteal expansion, while remodelling brings to the medullary the expansion of the long bones typical of aging. Besides, bone remodelling serves to build bone microarchitecture in order to meet mechanical needs, and it helps to repair micro-damage in the bone matrix and fractures healing. It also plays an important role in maintaining bone mineral turnover so that the levels of plasma calcium remain stable [27,28]. Osteoblasts and osteoclasts communicate with each other through direct cell-to-cell contact, cytokines, and extracellular matrix interactions: the balance of these interactions is critical for bone homeostasis and health. Since the 1980s, the identification of the receptor activator of nuclear factor-κB ligand (RANKL)/RANK/osteoprotegerin (OPG) signalling system has managed to clarify the role of osteoblasts and synovial fibroblasts in the regulation of osteoclasts’ differentiation and function [29]. RANKL/RANK signalling regulates the formation of multinucleated osteoclasts from their precursors, as well as their activation and survival in normal bone remodelling and in various pathologic conditions. RANKL is a member of the TNF–cytokine superfamily, which is expressed by osteoclastogenesis-supporting cells, including osteoblasts, osteocytes, and synovial fibroblasts, in response to osteoclastogenic factors, like parathyroid hormone (PTH), prostaglandin E2, and 1,25-dihydroxyvitamin D. It is also expressed by activated T-cells. The receptor for RANKL is RANK, a type I transmembrane protein, with high homology with CD40. It can be found on osteoclast precursor cells and mature osteoclasts [30]. RANKL/RANK signalling is essential for osteoclast formation, maturation, function, and survival. Many different intracellular pathways are activated by RANK-mediated protein kinase signalling [31]. The effect is the induction of the nuclear factor of activated T-cells (NFATc1), a master regulator of osteoclast differentiation, through the TNF receptor-associated factor 6 (TRAF6) and the c-Fos pathway [32]. NFATc1 regulates a number of osteoclastic-specific genes, like β3-integrin, cathepsin K, tartrate-resistant acid phosphatase (TRAP), calcitonin receptor, and osteoclast-associated receptor (OSCAR) [4]. The binding of RANKL to RANK is inhibited by the decoy receptor OPG, which is produced by osteoblasts. The RANKL/OPG ratio determines osteoclast activation and bone resorption [33].

The continuous modelling and remodelling process leads to a progressive bone mass increase, until it reaches a maximum value defined peak bone mass (PBM). PBM is achieved toward the end of the second decade of life, with 25% of PBM acquired during the 2 years of peak height velocity (HV). In fact, bone mass increase during puberty follows the peak velocity of growth at least 6–12 months later. Approximately 90% of PBM has been acquired by 18 years of age [34,35]. The exact age at which bone mass reaches its peak at various skeletal sites varies from 16–18 years, approximately, for vertebral column and femoral neck, and up to 35 years for the skull. After PBM is achieved, there is a slow but progressive decline in bone mass [36].

## 3. Paediatric Osteoporosis

Several pathological conditions, arising during the developmental age, can compromise the achievement of adequate PBM and/or cause bone loss. Furthermore, any condition that interferes with optimal PBM accrual may increase fracture risk later in life [26]. The acquisition of a higher PBM is the most important determinant of lifelong skeletal health. Conditions associated with lower bone mass and increased fracture risk in children and adolescents are reported in Table 1. Rare conditions with increased bone fragility are osteogenesis imperfecta (OI), idiopathic juvenile osteoporosis, and Turner syndrome. Cystic fibrosis, juvenile systemic lupus erythematosus (jSLE), juvenile idiopathic arthritis (JIA), inflammatory bowel disease (IBD), celiac disease, chronic renal failure, childhood cancers, and cerebral palsy can all be associated with reduced bone mass. Risk factors include malnutrition, increased metabolic requirements, intestinal malabsorption, low body weight, chronic inflammation with increased cytokine production, hypogonadism, immobilization, and the prolonged use of glucocorticoids (GC). Some medications, such as anticonvulsants and chemotherapeutic agents, and the prolonged use of proton pump inhibitors (PPI) and selective serotonin reuptake inhibitors (SSRI) can also impact bone mass acquisition negatively [26].

Osteoporosis has been traditionally considered as a geriatric disease, and its prevalence in the paediatric age has been widely underestimated. Only recently have paediatricians acknowledged its importance since some patients have genetic mutations that can affect bone metabolism, and many others are affected by chronic conditions, which can impact bone health. If not diagnosed and treated early enough, bone loss can proceed and affect PBM, with a relevant effect on fracture risk later in life. Primary osteoporosis is rare, mainly represented by OI, while conditions linked to low bone mass because of chronic disorders and drug administration are more frequent. Osteoporosis is generally defined by an altered microarchitecture of the bones (reduced number and thickness of bone trabeculae, increased porosity of cortical bone), and decreased BMC. According to the definition of the World Health Organization (WHO), osteoporosis is defined as a “systemic skeletal disease characterized by low bone mass and micro-architectural deterioration of bone tissue, with a consequent increase in bone fragility and susceptibility to fractures” [37]. In adults, osteoporosis is defined as a BMD of 2.5 or more standard deviations (SDs) below the young adult mean (a T-score of less than 2.5), while the Z-score should be used in interpreting dual-energy X-ray absorptiometry (DXA) results in children. The Z-score is the number of SDs below the age-matched mean. The reasons for this are, first of all, children have not yet achieved PBM. Second, DXA measures 2-dimensional aBMD, which is expressed as grams per square centimetre, and this method underestimates true vBMD in subjects with smaller bones. Third, many children with chronic illness have growth retardation and delayed puberty. Therefore, a correction should be made for height or height age [38,39,40]. According to the *Official Positions of the International Society for Clinical Densitometry (ISCD),* published in 2013, the diagnosis of osteoporosis should not be based on densitometric criteria alone. Paediatric osteoporosis can be defined in the presence of one or more vertebral fractures (VFs) in the absence of local disease or high-energy trauma, or in the case of a history of clinically significant fractures, specifically 2 or more long bone fractures occurring by age 10 or 3 or more long bone fractures at any age up to 19, plus a Z-score of BMD or BMC ≤ −2 (age- and gender-matched and adjusted for size in case of impaired growth) [41]. This definition allowed a shift from a BMD-centric definition of osteoporosis to a fracture-focused approach in the diagnosis and follow-up.

DXA is the most frequently used method of assessment of bone mass because of its availability, diagnostic accuracy, and because of its use of a low dose of radiation (5–6 mSv for the lumbar spine, hip, and whole body) [42]. DXA measures BMC and calculates areal BMD (aBMD) by dividing BMC by the area of the region scanned. The software of the major DXA manufacturers utilizes paediatric reference databases for children older than 5 years [43]. In children, the favourite sites of measurement are the lumbar spine (LS) and the whole body less head (WBLH) [44]. Newer technologies, such as quantitative computed tomography (QCT) and peripheral quantitative computed tomography (pQCT), are three-dimensional techniques that also use the attenuation of X-ray beams to evaluate BMC. Cortical and trabecular bone compartments vary in density, such that X-ray beams are attenuated to them in different ways; this method allows separate determinations of the trabecular and cortical volumetric BMD (vBMD) of the appendicular skeleton. In fact, while DXA measures two-dimensional aBMD (expressed as grams per square centimetre), QTC measures three-dimensional vBMD (expressed in grams per cubic centimetre) [45]. Bone information can also be obtained by magnetic resonance imaging (MRI), which is based on the quantity of marrow tissue present within the bone-free spaces of the trabecular compartments. Normal bone presents higher numbers and greater surfaces of trabeculae with narrower marrow spaces, while osteoporotic bone has lower numbers and decreased surfaces of trabeculae with wider marrow spaces. Evaluated bone districts are the calcaneus, the medial portion of the tibia, and the distal phalanges of the non-dominant hand or the metacarpus in children under 3 years of age [46]. Another technique is echography: it is based on the measurement of the degree of attenuation or the speed of the ultrasounds while they cross through the bone segment under examination. This technique provides not only quantitative but also qualitative information on the bone of the patient, and it can be integrated with DXA [45].

## 4. Secondary Osteoporosis in Children with Rheumatic Diseases

Secondary osteoporosis is now recognized as an important issue of clinical care in children with acute and chronic diseases. In these patients, the underlying disease’s pathogenesis and/or its treatment play a fundamental role in influencing bone health [47]. A great variety of conditions has been associated with paediatric secondary osteoporosis, the most common being compromised mobility, GCs-treated inflammatory diseases, Duchenne muscular dystrophy (DMD) and other neuromuscular disorders, leukaemia and other cancers, renal diseases (nephrotic syndrome), hypogonadism, and thalassemia [48]. In these patients, multiple factors contribute to the reduction of bone strength: these include elevated cytokines, immobility with decreased muscle mechanical loading on bone, nutritional deficiency, insufficient daily calcium and vitamin D intake, the use of toxic medications, poor growth, and pubertal delay. Each of these factors needs to be considered and managed to improve and optimize bone health in at-risk children [49].

The relation between chronic inflammatory diseases and altered bone metabolism in children has been extensively investigated. Systemic autoimmune disorders, such as IBD, jSLE, JIA, juvenile dermatomyositis (JDM), systemic vasculitis, and overlap syndromes, can affect the skeletal system, causing reduced BMD and a high risk of fragility fractures during childhood. Reduced BMD can be observed at all sites of the skeleton in children and adolescents with JIA, and up to 50% of adults with a history of JIA show reduced bone mass [50]. Lien et al. measured total body, LS, and femoral neck BMC and BMD in 105 adolescents with early-onset JIA, and with a mean disease duration of 14.2 years. They found that 41% of patients had low total-body BMC, which was related to a decrease of mean weight and height, reduced lean mass, disease duration, disease activity, and number of involved joints [51]. The achievement of about 50% of PBM occurs during puberty, thanks to increases in bone cortical thickness and trabecular mineralization. Therefore, low PBM may result from clinical conditions characterized by pubertal abnormalities [52]. In fact, pubertal delay is an additional insult to bone health in children with JIA, together with elevated inflammatory cytokines, GC use, low physical activity, altered body composition, and malnutrition with calcium and vitamin D deficiencies [53]. Therefore, bone mass accrual is suppressed in patients with JIA, leading to future bone fragility and an increased fracture risk [54]. Burnham et al. conducted a population study including 1939 JIA patients and about 200,000 controls. They showed a clinically significant increased risk of non-vertebral fractures (non-VFs) in the first group (6.7% in patients vs. 3.3% in controls) [55]. In juvenile arthritis, osteoporosis mainly develops at the femoral neck and radius compared to the lumbar spine [56]. Nevertheless, VFs are an important, under-recognized manifestation of secondary osteoporosis, especially in the polyarticular and systemic subtypes, and GC therapy is among the most potent risk factors for the development of both VF and non-VFs in these patients. Different studies demonstrated a 6–34% prevalence of anterior wedge or compression vertebral crashes in children with chronic inflammatory diseases [57,58]. The Canadian Steroid-Associated Osteoporosis in the Paediatric Population (STOPP) Consortium evaluated spine health among 134 children with rheumatic conditions within 30 days of starting GC therapy. They found VFs in 7% of patients; in the systemic JIA (sJIA) group, 2/22 children developed VFs (9%), while no VFs were observed in 28 children with other JIA subtypes. Children with incidental VFs had a greater decrease in LS BMD Z-scores in the first 6 months [59].

### 4.1. The Role of Inflammation

It is known by now that the skeletal and immune systems share many regulatory molecules, including cytokines, signalling proteins, receptors, and transcription factors: an interesting link called “osteoimmunology” [60]. The interest in this topic arose from the observation of increased osteoclast-mediated bone loss in inflammatory diseases [61]. Accumulating evidence underlines the role of a complex network of bone cells, T and B lymphocytes, pro-inflammatory cytokines (IL-17, IL-23, IL-1, IL-6, and TNF-α), and signalling pathways, including the RANKL/OPG and Wnt. Masi et al. reported elevated serum OPG levels and decreased levels of RANKL in 84 JIA patients, compared to healthy controls [62]. Similarly, an elevated RANKL/OPG ratio was described in 37 children with JDM at the time of diagnosis, compared to healthy controls [63]. Furthermore, nuclear factor-κB (NF-κB) and NFATc1 are both key regulators of the immune response and transcription factors for osteoclast differentiation [61]. Above all, the increased local and systemic production of pro-inflammatory cytokines plays a pivotal role in the uncoupling of osteoblast-mediated bone formation and osteoclast-mediated bone resorption, thus altering normal paediatric skeletal remodelling [64]. Therefore, it can be supposed that anti-cytokine biologic treatment may have a beneficial effect on BMD and fracture risk, beyond the inhibitory effect on disease activity.

Inflammatory cytokines enhance RANKL expression on synovial fibroblasts, osteoblasts, and osteocytes. They also down-regulate Wnt pro-osteogenic signalling by the induction of its antagonists, Dickkopf proteins (DKK1 and DKK2), soluble Frizzled-related proteins (sFRPs), and sclerostin [65,66].

TNF-α is a powerful inducer of osteoclastogenesis. In fact, it enhances the expression of OSCAR, involved in osteoclasts’ function and increases RANKL expression [67]. Some authors have reported that TNF-α and IL-6 stimulate osteoclast formation and bone erosion, even in a RANKL-independent way [68]. It also inhibits osteoblast activity by increasing DKK1 production and restraining some transcription factors involved in osteoblast differentiation, like RUNX family transcription factor 2/core-binding transcription factor 1 (Runx2/Cbfa1) and osterix [69,70]. Simonini et al. demonstrated increased bone mass after 1 year of anti-TNF treatment with etanercept in children with JIA [71]. Later, some authors reported that clinical control of disease activity by etanercept in patients with methotrexate (MTX)-refractory poly-JIA was associated with rapid catch-up growth and bone mineralization improvement. They also provided preliminary data that a beneficial etanercept effect may be related to the reduction of systemic IL-6 production and the enhancement of osteoblast activity, proven by increased OPG levels [72].

The TNF-α and IL-1β effects on bone loss are synergistic [73] since TNF-α induces IL-1β and its receptor in stromal cells, and IL-1β mediates TNFα-induced osteoclastogenesis: it up-regulates RANKL expression and prostaglandin E2 secretion and stimulates DKK1 and sclerostin activity [74].

On the other hand, IL-6 has shown to have an important role in the development of autoimmune diseases associated systemic osteoporosis. It interacts with IL-1β and TNF-α to support osteoclast activity through RANKL-mediated and Janus Kinase (JAK)-mediated pathways. It also plays a critical role in the generation of Th-17 cells. Finally, it acts through the stimulation of the hypothalamic–pituitary-–adrenal axis, leading to the secretion of steroid hormones contributing to bone loss [69]. De Benedetti et al. showed that prepubertal IL6-transgenic mice have delayed ossification, decreased cortical and trabecular bone, uncoupled osteoclast and osteoblast activities, reduced mineral apposition rates, and the impaired growth of skeletal segments. These results imply that chronic IL-6 overexpression in growing prepubertal animals can dissociate osteoclast and osteoblast functions, leading to the development of an osteopaenic phenotype [75].

It is known by now that RANKL is expressed in activated T-cells. The role of a group of osteoclastogenic helper T-cells, Th-17 cells, has been largely investigated. Th-17 cells’ differentiation is stimulated by the combination of IL-6 and TGF- β, leading to the production of pro-inflammatory cytokines, including IL-17, IL-22, and IL-23. Th17 cells play a pivotal role in host defence against extracellular pathogens and concurs to the development of many autoimmune diseases [76]. IL-17 induces RANKL on osteoclast-supporting mesenchymal, cells both directly and indirectly, through the up-regulation of other inflammatory cytokines (IL-1, IL-6, and TNF-α) [77,78]. On the other hand, Th-17 cells express RANKL on their surface. Cathepsin K is a lysosomal protease involved in osteoclast-mediated degradation of bone matrices. It is an interesting example of the link between bone and immunity since it induces Th-17 cells, through IL-6 and IL-23 in dendritic cells, contributing to autoimmune inflammation. [4]. On the contrary, Th-1 and Th-2 derived cytokines, including interferon-γ (IFN-γ), IL-10 and IL-4, have massive inhibitory effect on osteoclast differentiation [79]. Primarily, IFN-γ reduces osteoclastogenesis through ubiquitin–proteasome-mediated degradation of TRAF6 [80]. However, IFN-γ is not largely expressed in the joints of patients with rheumatoid arthritis, suggesting a minor role of Th-1 cells in the development of arthritis-related bone damage [3]. Similarly, CD4^+^CD25^+^ T regulatory (Treg) cells inhibit osteoclast differentiation through TGF-β, IL-4 and IL-10 production, and also via cell-to-cell contact mediated by cytotoxic T lymphocyte antigen 4 (CTLA-4), a negative regulator of T-cell activation [81]. It can be speculated that abatacept, a CTLA-4-Ig fusion protein, may exert an anti-erosive effect in patients with inflammatory arthritis [82]. Similarly, Zaiss et al. reported that Foxp3^+^ Treg cells protect against local and systemic bone loss in TNF-α induced arthritis in mice [83]. Nevertheless, Treg function is affected by the specific immunologic microenvironment, failing to suppress effector T-cell proliferation or cytokine production in joint fluid in patients with rheumatoid arthritis [83].

There are contradictory data about the role of B-cells and auto-antibodies on bone remodelling [84,85]. B-cells express RANKL and IL-6; they also differentiate into plasma cells which inhibit bone formation through DKK1 expression [86]. Anti-citrullinated protein antibodies (ACPA) currently represent an interesting marker for the diagnosis of JIA, allowing for the diagnosis of rheumatoid factor (RF)-positive polyarticular JIA and the identification of JIA patients with severe bone involvement [87]. It was originally proposed that ACPA could indirectly mediate bone loss through the induction of TNF-α production by macrophages [88]. Recently, it has been reported that ACPA also bind directly to citrullinated proteins on the surface of osteoclast precursors, thus stimulating osteoclastogenesis [89]. Figure 2 summarizes the role of inflammation on osteoclasts and osteoblasts activity.

### 4.2. The Role of Glucocorticoids

GCs represent the mainstream treatment for many chronic and rheumatic diseases in children. They are associated with different direct and indirect effects on the growth plate and on the developing skeleton, causing defective bone remodelling and increased fracture risk [90]. Both daily mean doses and cumulative doses of GCs have been associated with GC-induced osteoporosis: the first was mainly related to increased fracture risk [91,92], and the latter to reduced bone mass [93]. No truly safe dose of GCs was reported since the risk is described even with a 2.5–7.5 mg daily dose [94]. GC-induced osteoporosis is influenced by additional factors, like age, gender, hormonal status, pubertal stage, baseline BMD, and underlying disease [95]. Even individual differences in GC sensitivity play a role. The different expression of 11β-hydroxysteroid dehydrogenase (11β-HSD) enzyme on osteoblasts, which interconvert inactive and active cortisol, may exert a prereceptor modulation of GC activity [96]. In addition, the observed variability in the severity of GCs adverse effects, including bone loss, could be associated with genetic polymorphisms in the GC receptor gene [97]. GC-mediated osteoporosis is mainly characterized by decreased bone formation, with an additional early and transient increase in bone resorption. The final effect is increased bone turnover, with a negative remodelling balance, causing rapid bone loss [98]. The effects of GCs on bone remodelling occur in two phases. The first, early phase is characterized by increased cytokine expression (macrophage colony-stimulating factor (M-CSF) and RANKL), enhanced osteoclastogenesis, decreased osteoclast apoptosis, and bone resorption [99]. As a consequence, BMD rapidly decreases in the first 2 weeks after GC initiation, even if important bone loss only develops during the first 3–6 months of therapy [100]. This effect diminishes with time, being replaced by a second, chronic phase of decreased bone formation. GCs directly act on bone formation via the up-regulation of peroxisome proliferator-activated receptor gamma receptor 2 (PPARγ2), which favours the differentiation of pluripotent precursor cells to adipocytes instead of osteoblasts, resulting in decreased numbers of osteoblasts [101]. Furthermore, they cause increased expression of sclerostin, which inhibits Wnt/β-catenin signalling, leading to reduced osteoblast differentiation and increased osteoblast and osteocyte apoptosis [102]. Apart from direct bone effects, GCs affect bone health through several indirect mechanisms. They favour a negative calcium balance by decreasing intestinal calcium absorption and stimulating renal tubular calcium excretion in the urine; they interfere with the growth hormone (GH)–insulin growth factor (IGF1) axis, leading to altered growth plate chondrocyte function and to GC-induced myopathy; they also interfere with gonadotropin secretion, causing hypogonadism and delayed puberty [103]. In addition, GC-induced weight gain affects the biomechanical load influencing bone homeostasis [104]. The final combined effect of GCs on bone is to reduce BMD [105] and to alter bone microarchitecture, with a predilection for the trabecular-rich bone, even if cortical bone is not spared [106]. Reduced trabecular bone formation is associated with increased endocortical resorption. Therefore, LS, femoral neck, and distal radius present reduced BMD in children with JIA treated with GCs [57]. Harrington et al. evaluated 15 children who developed fragility VFs while on chronic GC therapy for an underlying rheumatic disease. A trans-iliac bone biopsy was performed and histomorphometric analysis of the bioptic samples was undertaken. Collected analysis demonstrated a significant decrease in trabecular and osteoid thickness, reduction of osteoblast surface, and increase in trabecular separation compared to age-matched normative data. The severity of the trabecular deficit was strongly associated with the GC dose and treatment duration. Higher body mass index (BMI) and lower height Z-score, which are systemic adverse effects of GCs, also correlated with histomorphometric parameters [107]. VFs are the clinical hallmark of paediatric GC-induced osteoporosis. They occur early during GC treatment, are frequently asymptomatic, and may remain undetected if routine monitoring is not performed [108]. The risk of fractures rapidly decreases after GC therapy cessation; nevertheless, it remains above the baseline [94]. Different studies have reported a 7% prevalence of VFs in children with autoimmune disorders within 30 days of GC initiation [59], a prevalence of 29–45% later in the treatment course, with up to a 33% incidence in the first years of GC administration [109]. Recently, Ward et al. evaluated the incidence and predictors of fragility osteoporotic fractures and the potential for spontaneous recovery over 6 years following GC initiation in 136 children with rheumatic disorders. The 6-year cumulative fracture incidence was 10.1% for non-VF and 16.3% for VF. A total of 63% of VFs occurred in the first 2 years, with a peak incidence at 12 months, and 84% of children had complete vertebral body reshaping. Incident VFs were predicted by increased disease activity and BMI Z-scores in the first 12 months and reduced LS BMD Z-scores in the first 6 months. Higher average daily GC doses, instead, predicted both incident VF and non-VF: every 0.5 mg/kg increase in average daily GC dose was associated with a 2.1-fold increased risk of VF and non-VF over 6 years [110]. Children have the sole, growth-mediated potential to restore normal vertebral body height after VFs. This is an interesting element of paediatric osteoporosis since spontaneous vertebral body reshaping may occur even in the absence of a specific therapy. The peripubertal period is a critical moment in determining whether a child has sufficient residual growth potential to allow this vertebral reshaping. However, it is known that older children with less residual growth potential, children with poor growth, and children with ongoing risk factors (GC use, poorly controlled underlying disease, severe vertebral collapse and immobilization) have less potential for spontaneous recovery [108].

### 4.3. The Role of Growth and Pubertal Delay

Chronic inflammatory diseases are often associated with growth delay, ranging from a minimal decrease in HV to severe short stature [111]. Growth disorders have a multifactorial pathogenesis, including the role of the chronic inflammatory state, long-term use of GCs, malnutrition and malabsorption, physical inactivity, and disorders of pubertal onset. Chronic inflammation is known to interfere with the function of the GH–IGF1 axis [112]. It also affects growth-plate homeostasis at a local level. Pro-inflammatory cytokines inhibit the proliferative activity of growth cartilage and reduce the rate of endochondral ossification by decreasing chondrocyte proliferation and stimulating proliferative cell apoptosis [113]. Prolonged exposure to pro-inflammatory cytokines has been associated with a restricted potential for recovery of chondrogenesis and with reduced longitudinal bone growth, thus explaining greater growth impairment in children with longer symptomatic and severe uncontrolled disease [114]. Patients with inflammatory diseases often need to receive long-term steroid therapy. GCs can alter pituitary GH pulsatile release, reduce IGF-1 and GH receptor expression by chondrocytes, and influence IGF-1 signalling at the growth plate [115]. They also act directly on the growth plate by reducing chondrocyte proliferation and differentiation and stimulating apoptosis [116]. Given the presence of GH–IGF-1 axis alterations in chronic inflammatory conditions, the role of recombinant human GH (rhGH) therapy in prepubertal children with chronic rheumatic diseases has been largely investigated. Simon et al. studied the effect of rhGH therapy on growth velocity and body composition in patients with JIA. They showed an increase in lean mass of 33% and in lumbar BMD of 36.6% [117]. In these patients, GH treatment was associated with fat-mass reduction and the significant increase and normalization of muscle cross-sectional area (CSA) and total bone at final height [118]. The role of pubertal age should also be considered since patients with JIA and other inflammatory diseases often present with pubertal delay or slow pubertal progression. Puberty is characterized by growth-plate enlargement, increased bone cortical thickness, and enhanced trabecular mineralization [52,53,54]. Therefore, puberty is crucial for PBM acquisition [119]. The international study *The Bone Mineral Density in Childhood Study* demonstrated that the age at onset of puberty is an important negative predictor of BMC and BMD in all skeletal sites in adulthood [120]. Some authors reported reduced BMD Z-score values in JIA patients with delayed puberty, with a strict relationship between peripubertal mineralization and the age of menarche [121].

### 4.4. The Role of Phisical Inactivity

Physical activity influences up to 17% of the variability in BMD [122]. Mechanical loading induces bone tissue strain, sensed by the osteocyte system: osteoblast and osteoclast activity is stimulated when tissue strain exceeds a given threshold in order to reinforce bone tissue at the site of bone strain [123]. Therefore, bone mass and geometry follow the development of body size and muscle force during the growing age [124]. Conversely, autoimmune inflammation is associated with myopathy and reduced lean mass in patients with rheumatic diseases, with negative effects on bone geometry. Inflammatory cytokines, especially TNF-α, inhibit myocyte differentiation, while stimulating muscular cell apoptosis and protein degradation [125]. Therefore, rheumatoid cachexia is an important known complication of inflammatory disorders [126]. In healthy children, moderate and vigorous activity leads to increased lean mass and bone mass [127], and high-impact activity has an anabolic effect on the growing skeleton, particularly in prepubertal children and in early puberty [128]. JIA patients, instead, have reduced physical activity, due to bone pain, functional limitation, and lifestyle [129]. Some studies reported reduced muscle strength, reduced muscle CSA measured by pQCT, and significant bone geometry abnormalities in JIA patients. Consequently, decreased skeletal size, reduced bone strength and altered trabecular and cortical bone density have been described, especially in children with greater disease activity [125,130].

## 5. Therapeutic Interventions

### 5.1. Bisphosphonates (BPs)

Despite the wide use of bisphosphonate (BP) therapy in adults, significant controversy remains regarding its use in children [131]. BPs are considered as a rescue treatment for children with symptomatic osteoporosis, presenting with bone pain, recurrent fragility fractures, and VFs, and a lack of potential for unassisted recovery [132,133]. The use of BPs presents further indications in early puberty in order to favour achievement of PBM in this phase. Therefore, BP use can be considered for pubertal patients with risk factors, or with BMD Z-score ≤ −2.5 SDs with a downward trend confirmed on at least two separate occasions, 1 year apart. They can also be considered in the case of a BMD Z-score ≤ −3 SD with a declining trajectory on at least on two separate occasions, 12 months apart, even if active risk factors are not present [134]. The use of BPs for primary fracture prevention in GC-treated children is not recommended [135]. Finally, BPs should only be used after the optimization of vitamin D and calcium intake, physical therapies to maximize mobility, and GH treatment in the case of puberty disorders or hypogonadism [49]. In any case, with the exception of OI, BPs are used off-label in children, so informed consent must be obtained. Second- and third-generation BPs are mainly used in paediatric osteoporosis. Table 2 summarizes the most-commonly used BPs in children.

BPs are pyrophosphate-derived molecules in which a carbon atom has replaced an oxygen atom. They are chemically stable analogues of inorganic pyrophosphate, with a core phosphate–carbon–phosphate (P–C–P) unit, which is characterized by a strong binding affinity toward hydroxyapatite [136]. BPs can be classified into nonnitrogen-containing and nitrogen-containing BPs. The former group, the first-generation BPs (etidronate, clodronate, and tiludronate), inhibit mitochondrial adenosine triphosphate (ATP) translocases and induce osteoclast apoptosis [137]. On the other hand, nitrogen-containing BPs have higher antiresorption activity since they have great affinity for bone mineral, and they strongly inhibit osteoclast function. This group of BPs targets osteoclast farnesyl pyrophosphate synthase (FPP), a prenyl transferase involved in the mevalonate pathway, thus blocking protein prenylation and inhibiting bone resorption. It comprises pamidronate, alendronate, risedronate, ibandronate, and zoledronate [138]. BPs are retained in the skeleton and slowly released from bone, presumably after the resumption of bone remodelling at previously exposed sites. Finally, they are excreted by urine for a very long time. Indeed, some authors reported that pamidronate could be detected in urine samples up to 8 years after the cessation of daily oral treatment in 7 paediatric patients with severe osteoporosis [139].

Studies in children with OI have reported that BP treatment increases bone mass and reduces bone pain and fracture without major adverse effects [140]. Histomorphometric analysis showed a reduction of bone remodelling and resorption after BP therapy in children with neither a reduction in bone growth, nor trabecular nor periosteal bone formation. They also reported increases in both the trabecular number of metaphyseal bone and the cortical thickness. The consequence of BP administration to the growing child is the development of bone growth and modelling, resulting in a significant increase in bone mass and strength [141]. Some randomized control trials (RCTs) reported an improvement in BMD using different intravenous and oral BPs in a number of conditions associated with a high risk of osteoporosis, including JIA [142], Crohn disease [143] and other various GC-treated inflammatory disorders [144]. However, they mostly have low sample sizes and short follow-up durations, and they do not assess the outcome on fracture rates. Many case-control studies, while heterogeneous, have reported a positive effect of BP treatment in children with inflammatory conditions [145,146]. Thornton et al. evaluated the available evidence for the efficacy and safety of BPs in children with JIA-associated low BMD and fragility fractures. In all the collected studies, treatment with BPs allowed a mean percentage increase in spine BMD, ranging from 4.5% to 19.1%, compared with baseline. The most commonly described side effect was a transient, flu-like reaction, usually occurring after the first intravenous administration [147]. This self-limiting, acute-phase response develops with flu-like symptoms (bone pain, myalgia, fever, nausea, and vomiting) in up to 80% of patients, within 24–48 h after the first infusion. These symptoms simply resolve with analgesia and fluids [148]. BP-induced hypocalcaemia can also occur, even if severe symptomatic hypocalcaemia is rare. Vitamin D deficiency, prolonged GC use, subclinical hypoparathyroidism, and renal insufficiency are contributing risk factors [149]. Calcitriol administration for 3 days after the first dose may reduce the severity of these effects. Hypophosphatemia has also been described, but routine phosphate supplementation is not recommended [49]. The most commonly reported side effects for oral BPs are gastrointestinal symptoms and erosive esophagitis [150]. Other adverse events, such as BP-related, atypical, sub-trochanteric, femoral fractures; osteonecrosis of the jaw (ONJ); iritis; and acute renal failure, are extremely rare [151]. It is important to rehydrate the child prior to therapy administration in order to limit the possibility of BP-induced renal damage. The American College of Rheumatology recommends a referral for dental care before starting BP therapy to minimize the risk of ONJ. An annual or biannual dental review while on BPs is also advisable [50]. Furthermore, the teratogenicity of BPs has been investigated in animal models: these molecules have been shown to pass the placenta and to induce an increase in the number of diaphyseal bone trabeculae and the shortening of the diaphysis in the offspring [152]. In a small number of reports, no significant effects on the foetus have been reported when BPs have been used prior to conception [153]. Nevertheless, postmenarche girls should have a pregnancy test prior to BP administration and avoid pregnancy for 12 months after that [49].

### 5.2. Calcium and Vitamin D Supplementation

Vitamin D and calcium deficiency is quite common in chronic rheumatic diseases [154], and GC therapy further interferes with the calcium–phosphate balance since it reduces calcium resorption and causes secondary hyperparathyroidism [155]. Besides regulation of calcium homeostasis, 1,25-hydroxy-vitamin D has multiple antiproliferative, immunomodulatory, and anti-inflammatory properties. It suppresses Th-1 and Th-17 cells by decreasing Il-6, IL-12, and IL-23 secretion [156]; it promotes Th-2 cells by stimulating IL-4 and IL-5 production [157]; it also down-regulates B-cell proliferation, plasma cell differentiation, and immunoglobulin production [158]. Therefore, vitamin D deficiency can play a role in the pathogenesis of autoimmune diseases [159], and vitamin D replacement may counteract the negative effect of inflammation on bone health. The optimal vitamin D serum level is still controversial. It is recommended that, to ensure proper vitamin D intake, one should maintain plasmatic 25-hydroxy-vitamin D levels higher than 50 nmol/l (20 ng/dl) [134]. Some authors have proposed a doubled recommended daily allowance (RDA) in children with JIA and other rheumatic conditions, as compared to healthy ones [160]. Vitamin D supplementations of up to 2000 IU per day appear to be safe and well-tolerated in children with chronic diseases [161]. In paediatric patients with low BMD or osteoporosis, the optimization of calcium dietary intake is preferred over therapeutic supplementation. Table 3 summarizes the daily calcium and vitamin D requirements in children.

### 5.3. Physical Activity and Muscle Training

According to “mechanostat theory”, during the paediatric age, bone development is driven by two mechanical challenges which induce bone tissue strain: the increase in bone length and in muscle forces and bone length [162]. Therefore, muscle loss can explain a significant amount of bone loss in JIA children due to reduced mobility and lifestyle changes [125]. Fazaa et al. conducted a cross-sectional study in order to measure physical activity levels in 55 children with JIA compared to 55 healthy controls matched by gender and age. They reported that many JIA patients spent the day sitting or sleeping and presented significantly lower levels of physical activity, compared to healthy children. This was mainly associated with the systemic or polyarticular form and with a more sever course of disease [129]. Pain seems to be the most important barrier to physical activity, as reported by patients and their parents, together with joint stiffness, muscular weakness, joint swelling and deformities, and constitutional symptoms. It has been shown that patients with JIA present reduced muscle strength and total body BMD compared to controls and that vigorous physical activity is able to improve muscle strength and cardiorespiratory performances in these patients [163]. Furthermore, physical activity itself can be a useful strategy to reduce and manage stiffness and pain [164], alleviate fatigue, and improve poor health-related quality of life [165]. Growing evidence from the literature also suggests that physical exercise induces anti-inflammatory effects in different tissues, including adipose tissue, vessels, and muscles [166]. Interestingly, soluble TNF receptors, which inhibit TNF-α, and IL-1 receptor antagonists, are produced during exercise and remain elevated even after its cessation. In addition, muscular macrophages shift to the anti-inflammatory subtype 2 (M2), thus affecting the expression of pro-inflammatory cytokines [167]. These observations reinforce the concept that effective personal strategies should be developed to foster physical activity in children with JIA. An important goal of physical activity in these JIA patients is to improve aerobic capacity, muscular power, and BMD. Increasing levels of physical activity can also alleviate pain, fatigue, muscle weakness, and poor health-related quality of life. Many studies evaluated the efficacy and safety of different exercise interventions in children with inflammatory arthritis [168,169]. Some authors reported increased femoral neck BMD in female adolescents after 15 months of resistance training [170]. High-impact activities and weightlifting should be introduced if tolerated since they have proven to be the most effective in triggering bone modelling and remodelling in growing subjects [171]. Recently, it was also reported that core stability exercise, in addition to conventional physical therapy, improves DXA-measured bone mineralization of the LS and femoral neck and enhances functional capacity in children with polyarticular JIA [172].

## 6. Conclusions

Secondary osteoporosis is an important comorbidity in children with rheumatic diseases. The inflammatory state often affects the skeletal system, leading to osteoporosis, fractures, and to an increased risk of reduced BMD in adulthood. In fact, in chronic arthritis, osteoimmune interactions cause skeletal alterations, in the form of bone erosions, periarticular osteopenia near the inflamed joint, and impaired systemic bone accrual. JIA and other inflammatory diseases are also characterized by bone pain, muscular weakness, and limited weight-bearing activity, with a loss of positive muscle–bone trophic effect [173]. Other contributing factors are nutritional deficiency, malabsorption, and osteotoxic drug employment. Furthermore, growth and pubertal development can be impaired in children with chronic diseases, leading to altered cartilage–bone homeostasis. The alterations of the musculoskeletal system often persist even after remission of the underlying disease, thus representing an important secondary outcome variable in the follow-up of these patients. Therefore, a standardized diagnostic approach to the musculoskeletal system, including prophylaxis and therapy, is mandatory in all children with JIA, especially those who do not achieve rapid disease remission and who require chronic GC treatment [50]. A key element in the management of secondary osteoporosis is the control of the underlying disease activity. Overall, available data have reported the positive effects of TNF-α blockers on BMD, leading to a reduced risk of systemic osteoporosis in JIA patients. However, the actual effect on fracture risk still needs to be determined [174]. Along with biologic therapy, GC sparing also plays an important role. BPs are the chosen treatment for symptomatic childhood osteoporosis, following vitamin D and calcium intake optimization and lifestyle interventions. Future studies are required to assess precise indications about initiation and duration of therapy. Furthermore, clinical research is needed to find specific prevision markers of bone health alterations in children with chronic diseases, to recognize patients with the possibility for spontaneous recovery, and to guide laboratory and radiological follow up. The diagnostic and prognostic role of bone turnover markers must also be defined in the paediatric age [175].

## Figures and Tables

**Figure 1 ijms-23-05725-f001:**
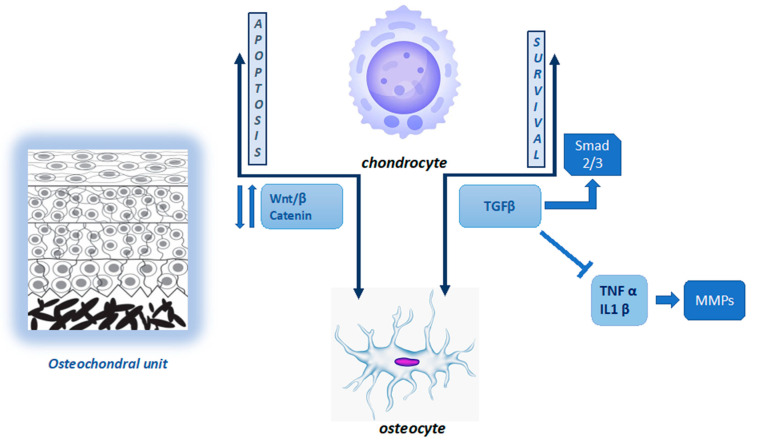
Cartilage–bone interaction in the osteochondral unit.

**Figure 2 ijms-23-05725-f002:**
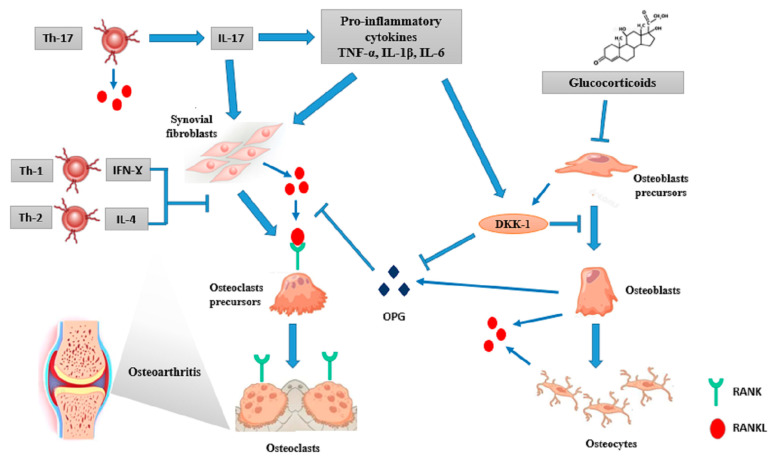
The role of inflammation on bone metabolism.

**Table 1 ijms-23-05725-t001:** Causes of reduced bone mineral density in children.

*Primary causes:* - Juvenile idiopathic osteoporosis *Genetic diseases:* - Gaucher disease - Wilson disease - Mucopolysaccharidoses - Glycogen storage disorders - Galactosemia - Turner syndrome - Klinefelter syndrome - Down syndrome - Noonan syndrome - Osteogenesis imperfecta - Marfan syndrome - Ehlers-Danlos syndrome - Menkes disease - Hypophoshatemic rickets *Endocrine diseases:* - Hypogonadotropic hypogonadism - Delayed puberty - Growth hormone deficiency - Hyperthyroidism - Type 1 diabetes mellitus - Cushing’s syndrome - Primary hyperparathyroidism - Acromegaly and hyperprolactinemia - McCune-Albright’s syndrome - Panhypopituitarism *Chronic diseases:* - Cystic fibrosis - Chronic kidney disease - Inflammatory bowel disease - Celiac disease - Juvenile idiopathic arthritis - Systemic lupus erythematosus - Juvenile dermatomyositis - Congenital heart disease *Hematological diseases:* - Thalassemia - Sickle cell disease - Haemophilia *Drugs:* - Corticosteroids - Anticonvulsant - Anticoagulants - Chemotherapy (methotrexate, cyclosporine) - Antiretroviral therapy - Gonadotropin-releasing hormone analogue *Tumors:* - Lymphoblastic leukemia and lymphoma - Neuroblastoma *Nutritional causes:* - Malnutrition - Diet without milk and derivates - Nervous anorexia and binge eating disorder - Vegetarian diet - Total parenteral nutrition - Obesity *Reduced physical activity and muscoskeletal disease:* - Cerebral palsy - Duchenne muscular dystrophy - Progressive spinal amyotrophy - Spinal neural tube defects - Poliomyelitis - Prolonged immobilization

**Table 2 ijms-23-05725-t002:** Main bisphosphonates used in the paediatric age.

Drug	Route of Administration	Dose	Relative Potency
**Etidronate** **(I generation)**	Oral Intravenous	p.o.:5–40 mg/kg per day iv: 400 mg per day for 2 wk, every 3 mo	1
**Pamidronate** **(II generation)**	Intravenous	<1 year: 0. 5 mg/kg every 2 mo 1–2 years: 0.25–0. 5 mg/kg/day 3 days every 3 mo 2–3 years: 0.375–0.75 mg/kg/day 3 days every 3 mo >3 years: 0.5–1 mg/kg/day 3 days every 4 mo Maximum dose: 60 mg/dose and 11.5 mg/kg/yr	100
**Alendronate** **(II generation)**	Oral	1–2 mg/kg/wk <40 kg: 5 mg/day or 35 mg/wk >40 kg: 10 mg/day or 70 mg/wk Maximum dose: 70 mg/wk	100–1000
**Neridronate** **(III generation)**	Intravenous	1–2 mg/kg/day every 3–4 mo	100
**Zolendronate** **(III generation)**	Intravenous	0.0125–0.05 mg/kg every 6–12 mo Mmaximum dose: 4 mg)	>10.000
**Risendronate** **(III generation)**	Oral	<40 kg: 15 mg/wk >40 kg: 30 mg/wk Maximum dose: 30 mg/wk	1.000–10.000

**Abbreviations**: p.o. (per os); iv (intravenous); wk (week); mo (month); yr (year).

**Table 3 ijms-23-05725-t003:** Calcium and vitamin D recommended daily intake for healthy children and for children with rheumatic diseases (adapted from Vojinovic and Cimaz [160] and Galindo-Zavala et al. [134]).

Age	Calcium (mg/Day)	Vitamin D			
		IOM Recommendations for Healthy Children		IOM Recommendations for Healthy Children at Risk of Vitamin D Deficiency	Proposed Recommendations for Children with Rheumatic Diseases
		EAR (IU/Day)	RDA (IU/Day)	IU/Day	IU/Day
**0–6 mo**	200	400	-	400–1000	1000
**6–12 mo**	260	400	-	400–1000	1500–2000
**1–3 yrs**	700	400	600	600–1000	2000
**4–8 yrs**	1000	400	600	600–1000	2000
**Boys** **9–18 yrs**	1300	400	600	600–1000	2000
**Girls** **9–18 yrs**	1300	400	600	400–2000	2000–3000

**Abbreviations:** mo (months); yrs (years); IOM (Institute of Medicine); EAR (estimated average requirement); RDA (recommended daily allowance); IU (international unit).

## Data Availability

Not applicable.

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
