# Peer review of "Bone Health in Children with Rheumatic Disorders: Focus on Molecular Mechanisms, Diagnosis, and Management"

_ijms, 2022, doi:10.3390/ijms23105725_

Round 1

Reviewer 1 Report

Authors have performed a detailed review of bone health of children. However, I did not identify “new insights into molecular mechanisms”. I think author’s descriptions are biased towards classical knowledge.

  1. The authors cite many articles (179), about 110 of which are older than 10 years. Although historical data is important, in my opinion the readers expect rather newer information. I think author’s descriptions are biased towards classical knowledge. I suggest selecting a new topics and description of the subject in relation to the latest reports and omission of historical/classical data.
  2. Fig 1. illustrated cartilage-bone interaction. Fig 2. Illustrated the effect of cytokines on bone metabolism. But I think that these knowledges are classical, that had been reported more than 5 years ago. Furthermore, these results are not mainly from the study that target children.
  3. Among the topics that authors dealt with, molecular biology of mechanical stress is relatively new field. Discussion about relationship between pediatric bone health and molecules such as Piezo2 will be attractive.

Author Response

Thank you for your report. We respond point by point to your comments:

Authors have performed a detailed review of bone health of children. However, I did not identify “new insights into molecular mechanisms”. I think author’s descriptions are biased towards classical knowledge.

Thank you for this useful remark. New information have been added in the text. Furthermore, we modified the title of the review, so that it is accordance with the topic of the work: “Bone health in children with rheumatic disorders: focus on molecular mechanisms, diagnosis and management”.

The authors cite many articles (179), about 110 of which are older than 10 years. Although historical data is important, in my opinion the readers expect rather newer information. I think author’s descriptions are biased towards classical knowledge. I suggest selecting a new topics and description of the subject in relation to the latest reports and omission of historical/classical data.

Thank you for this important point. We added some new information in the test and removed excessive historical data, as you suggested.

Fig 1. illustrated cartilage-bone interaction. Fig 2. Illustrated the effect of cytokines on bone metabolism. But I think that these knowledges are classical, that had been reported more than 5 years ago. Furthermore, these results are not mainly from the study that target children.

Thank you for this tip. Even if these figures are based on classical knowledges, we believe that they can be useful to resume and simplify the complex mechanisms described in the text.

Among the topics that authors dealt with, molecular biology of mechanical stress is relatively new field. Discussion about relationship between pediatric bone health and molecules such as Piezo2 will be attractive.

Thank you for your interesting suggestion. We included in the text information about Piezo function in mechanotrasduction. However, these data derive only from studies in adults, since we were not able to found studies about the pediatric population. 

Reviewer 2 Report

Please find my comments in the document

Author Response

Many thanks. We answer point-by-point to your suggestions.

This review aims to make a synthesis of the literature for rheumatic disorders in children and the impacts of bone heath and the molecular mechanisms involved. The review is well documented and the figures are of good quality.

I am not sure that the title is total accordance with the review.

The plan of the review does not seem to me the most appropriate. The sequence of the different parts is not always logical.

Moreover, the notion of rheumatic disorders comes in late. It is difficult to determine if osteoporosis is inherent to the pathology or to the treatments or to both?

I strongly suggest to modify the plan and give a more logical way.

We thank you very much for these useful remarks. As you suggested, the title has been changed and the plane of the work has been modified to provide a more logical sequence. Also the notion of rheumatic disorders has been reported earlier. Both underlying disease and its treatment are involved in the patogenesis of secondary osteoporosis. However, it is difficult to distinguish their contribution in the single case.  

Comments:

JOINT HOMEOSTASIS :

If the topic is bone, it is difficult to understand why the authors have included joint. Even if the authors deal with the subchondral bone, I’m not sure that the first part in really useful. I suggest to delete it or strongly reinforce the link between joint and bone. Especially since many bones do not have a joint.

We thank you very much for this useful remark. We deleted the first part and underlined the concept of link between bone and joint.

This first comment leads me to a second one, does the journal target all the bones or only the bearing and locomotor bones?

Thank you for this interesting point. The journal target all bones, since secondary osteoporosis is a systemic disease. Bone metabolism alterations in rheumatic disorders vary from local juxta-articular osteopenia to systemic osteoporosis due to chronic inflammation, physical inactivity, nutritional deficiency and glucocorticoid treatment.

BONE TISSUE REMODELING AND PEAK BONE MASS:

In my opinion, this paragraph should include a section on bone growth in the different physiopathological contexts. This is different from what is dealt with afterwards (e.g. bone remodeling in pathologies).

Thank you for your remark. The main of our work is not to talk about bone growth, but it is to talk about bone remodeling and bone metabolism in physiological conditions and in rheumatic diseases.

PEDIATRIC OSTEOPOROSIS:

The transition with this art has to be improved. Why directly osteoporosis and no other bone diseases?

We thank you for this important point. We talked about osteoporosis because the topic of the review is bone tissue metabolism and its alterations in pathological conditions. Indeed, a brief explanation of the mechanisms of physiological bone remodeling is followed by an insight on it alterations in pathological conditions, expecially in rheumatic diseases.

OSTEOPOROSIS IN RHEUMATIC DISEASES:

Why deal with this topic so late and after osteoporosis in children? this is not the subject of the review?

Thank you for this remark. We talked about osteoporosis in rheumatic diseases early in the text as you suggested. Prior we wanted to provide brief information about bone metabolism in children and about the diagnostic criteria of osteoporosis in the pediatric age.

THE ROLE OF INFLAMMATION :

Probably one of the more important part, it should be described much earlier.

Thank you for your suggestion. The sequence of the different parts of the review has been modified and some pharagraphs have been removed, so that the pharagraph “the role of inflammation” appears early in the text.

In the treatment part, the physical activities as treatment is poorly described. Please develop.

Thank you for this useful tip. We developed this topic.

Round 2

Reviewer 1 Report

 Authors added new informations, and I think this review is more engaging.  The revised version can be accepted in the present form.